# Understanding Leisure Centre-Based Physical Activity after Physical Activity Referral: Evidence from Scheme Participants and Completers in Northumberland UK

**DOI:** 10.3390/ijerph18062957

**Published:** 2021-03-13

**Authors:** Jordan Bell, Lis Neubeck, Kai Jin, Paul Kelly, Coral L. Hanson

**Affiliations:** 1School of Health and Social Care, Edinburgh Napier University, Edinburgh EH11 4DN, UK; l.neubeck@napier.ac.uk (L.N.); c.hanson@napier.ac.uk (C.L.H.); 2Charles Perkins Centre, University of Sydney, Sydney 2006, Australia; 3Centre for Medical Informatics, University of Edinburgh, Edinburgh EH16 4UX, UK; kai.jin@ed.ac.uk; 4Physical Activity for Health Research Centre, Institute for Sport, Physical Education and Health Sciences, University of Edinburgh, Edinburgh EH8 8AQ, UK; p.kelly@ed.ac.uk

**Keywords:** physical activity, exercise referral, public health, adherence

## Abstract

Physical activity referral schemes (PARS) are a popular physical activity (PA) intervention in the UK. Little is known about the type, intensity and duration of PA undertaken during and post PARS. We calculated weekly leisure centre-based moderate/vigorous PA for PARS participants (n = 448) and PARS completers (n = 746) in Northumberland, UK, between March 2019–February 2020 using administrative data. We categorised activity levels (<30 min/week, 30–149 min/week and ≥150 min/week) and used ordinal regression to examine predictors for activity category achieved. PARS participants took part in a median of 57.0 min (IQR 26.0–90.0) and PARS completers a median of 68.0 min (IQR 42.0–100.0) moderate/vigorous leisure centre-based PA per week. Being a PARS completer (OR: 2.14, 95% CI: 1.61–2.82) was a positive predictor of achieving a higher level of physical activity category compared to PARS participants. Female PARS participants were less likely (OR: 0.65, 95% CI: 0.43–0.97) to achieve ≥30 min of moderate/vigorous LCPA per week compared to male PARS participants. PARS participants achieved 38.0% and PARS completers 45.3% of the World Health Organisation recommended ≥150 min of moderate/vigorous weekly PA through leisure centre use. Strategies integrated within PARS to promote PA outside of leisure centre-based activity may help participants achieve PA guidelines.

## 1. Introduction

Insufficient levels of PA are one of the major risk factors for death worldwide [1]. There is strong evidence to show that physical inactivity increases the risk of coronary heart disease [2], type 2 diabetes [3] and some cancers [4,5]. It is estimated to cause 9% of premature global mortality [6] and in the UK, directly contributes to one in ten premature deaths from cardiovascular disease and one in six deaths from any cause [6]. Physical inactivity was estimated to cost the National Health Service (NHS) GBP 900 million in 2015, rising to GBP 1.2 billion in 2017 [7]. To mitigate mortality from inactivity, the World Health Organisation (WHO) recommends that adults undertake at least 150 min of moderate activity per week [8] but in 2019, 36.8% of adults aged over 18 years in England failed to meet these guidelines and 24.6% of these did less than 30 min per week [9]. These data provide a powerful case for exploring interventions that increase PA for those who are not sufficiently active to benefit health.

Physical activity referral schemes (PARS), also known as exercise referral schemes, have proven a popular PA intervention since their inception in the 1990s [10]. In the UK, they involve patients with a range of medical conditions being referred by healthcare professionals (HCP’s) to leisure providers for a programme of supervised PA [10]. UK PARS typically last between 8–26 weeks and usually consist of a PA prescription of 1–2 sessions per week [11]. While evidence supports positive short-term changes in PA levels, the benefit of PARS in increasing long-term PA levels is equivocal [12,13]. This is perhaps not surprising given the reported short-term nature of many PARS [14,15,16,17,18] and a lack of studies that examine what happens to participants following completion of such schemes. There is some consensus that longer-length schemes (20+ weeks) are more beneficial in improving health outcomes and behaviours [11]. Despite this, current UK guidance recommends schemes last for at least 12 weeks [19] and policy guidance surrounding best practice is lacking [20].

Data heterogeneity limits understanding of PARS effectiveness both at individual scheme level [10] and in systematic reviews [12,13]. A particular criticism is that there is little understanding of the type, intensity and duration of physical activity undertaken by PARS participants during schemes [21,22]. Where studies do report the effect of PARS on PA levels, it tends to be using pre- and post-scheme self-reported measures [14,23,24,25,26,27,28,29,30]. There have also been limited studies exploring PA habits following PARS. Most studies consist of self-report measures assessing PA levels 6–9 months post PARS completion [13]. A criticism of this is a limited understanding of the type of PA undertaken, and from a leisure provider perspective, there is an interest in continued use of leisure facilities post PARS. Therefore, studies are required that examine actual PA undertaken during and post PARS. This study aims to address this using a novel approach taking routine collected data via indirect observation to examine:

(1) The type and level of leisure centre-based activity undertaken in the short-term (up to 24 weeks) by participants in the Northumberland PARS.

(2) Longer-term (up to one year) levels of leisure centre-based activity undertaken by users who have completed PARS.

## 2. Materials and Methods

This study examined the contribution of local authority leisure centre usage to PA in PARS participants and PARS completers over a 12-month period in Northumberland, UK. We performed a retrospective analysis of local authority leisure centre usage for PARS participants and PARS completers by examining extracted anonymised PARS usage data from Active Northumberland, a charitable leisure trust that manages Northumberland local authority leisure facilities and provides the Northumberland PARS.

### 2.1. Context

Primary or secondary healthcare professionals could make referrals to the Northumberland PARS, which the local leisure trust provided at nine local authority leisure sites in Northumberland. We have previously reported a detailed description of the scheme process [26,30]. In brief, participants attended three consultations at the leisure site where they were referred to take part in the PARS (pre-scheme, after 12 weeks and post-scheme after 24 weeks) and were encouraged to attend two supervised PA sessions per week for a 24-week period. The Northumberland PARS is a rolling programme with some participants starting and finishing the scheme each week. Those who completed the scheme were encouraged to continue to attend leisure centre activities. We created two classifications of participants (PARS participants and PARS completers). We defined PARS participants as those undertaking the 24-week scheme at the point of data extraction and PARS completers as those who had finished the 24-week scheme but used the leisure centres during the data period examined (1 March 2019–29 February 2020). PARS participants may have attended for different periods of time up to a maximum of 24 weeks at the point of data extract. PARS completers had already had the opportunity to attend for at least the 24 week scheme duration. Those who had completed the PARS more than 28 weeks prior to the data extract had the opportunity to attend for the whole 52 week period, while, others may have had the opportunity to attend for between 25–52 weeks, dependent on the date that they completed the scheme. In our analysis we have accounted for the potentially different length of attendance for each individual. Only participants who had attended at least one activity during the data period were included in the analysis.

Sessions were available at a range of days/times and during their initial consultation, participants chose what days they would like to attend. Sessions were group based but tailored to individuals, with different activities available (gym, circuit classes, racquet sports and swimming/aqua aerobics). After completion of the PARS, participants had the option to attend structured exit route sessions (exclusively for PARS completers) and/or any other programmed leisure centre activities. Structured exit route sessions were also group based with different activities available (gym, circuit classes, racquet sports and aqua aerobics).

### 2.2. Study Setting and Dataset

All PARS participants were registered to use the leisure centre via the front desk system (FDS), Gladstone MRM (Gladstone Ltd., Oxford, UK), after attending their initial consultation and prior to attending their first PA session. The system recorded certain socio-demographic details (age, sex, and postcode) but not ethnicity or disabilities. The leisure trust issued every PARS participant with a swipe card, which they presented at each attendance to record activities or used for online booking. At the start of the PARS, the trust allocated participants a PARS subscription in the FDS lasting 24 weeks before automatically ending. After completion of the 24-week consultation, administration staff manually added a PARS completer subscription. The system was therefore able to provide objective, detailed user information about type, date and length of activities undertaken during and after scheme participation. Prior to data transfer, the leisure trust replaced FDS identification numbers with anonymised study identification numbers and used look-up tables [31] to classify PARS participant and completer postcodes by index of multiple deprivation (IMD) quintile (defining area level social economic status, with quintile one being the most deprived group and quintile five the least deprived group) [32].

### 2.3. Physical Activity Classification

Using previously established methods [33], we recorded the type and duration of activity available for PARS participants and completers prior to extraction. We allocated a Metabolic Equivalent of Task (MET) level for every activity using the Compendium of Physical Activities [34] and classified activities as light (<3 METs), moderate (3.0–5.9 METs) or vigorous (≥6 METs) intensity [35]. For example, we allocated a PARS group exercise class a MET value of 5.0 METs and classified it as moderate activity. With the exception of gym and swimming, activity duration was determined based on the timetabled duration.

At one leisure site, the PARS used the Technogym MyWellness System (Technogym S.p.A, Cesena, Italy) to record activities. This system recorded the amount of time each participant spent using cardiovascular and strength machines. Leisure trust staff routinely created a specific PARS participant group within the MyWellness System allowing for analysis of PARS participant workout data as part of scheme processes. Median workout time for PARS participants (n = 80) was 37 (interquartile range (IQR) 20–48.75) min. We therefore assigned an activity duration of 37 min for gym sessions. This included specific gym-based data for both PARS participants and PARS completers, but the groups were not differentiated within the MyWellness system. We were unable to measure swimming duration objectively; but as in our previous study, a 30-min workout time was applied based on estimates from trust staff [33]. Leisure trust staff provided the data extraction on a Microsoft Excel spreadsheet (Microsoft Corporation, Bellevue, Washington, DC, USA) with integrated METs values, intensity classification and duration for each activity.

### 2.4. Data Analysis

Our analysis included data for all users that had a PARS participant or PARS completer subscription on the data-extract date (1 March 2020) and had used any Northumberland leisure centre during the data extract period (1 March 2019–29 February 2020). Within the extract, there were no exclusions to the analysis. The final extract contained subscription type (PARS participant or PARS completer), sex, 10-year age group and IMD quintile. It also included individual usage data (date, duration, intensity level and type of activity undertaken for every attendance). We grouped individual activities into five main activity areas (PARS sessions, fitness classes, gym, swimming, and other activities such as badminton or 5-a-side football).

As described in detail in our previous work, we calculated the total number of attendances at light, moderate and vigorous activities, and the total duration of activities in each intensity category during the data extract period. We created a data field for the maximum number of weeks usage in the 12-month period and calculated the total weekly moderate/vigorous leisure centre-based PA (LCPA) per user. Specifically LCPA was adjusted for maximum weeks usage [33].
$$\mathrm{Total}\;\mathrm{weekly}\;\mathrm{LCPA}\;=\;\frac{\mathrm{total}\;\mathrm{duration}\;\mathrm{of}\;\mathrm{moderate}\;\mathrm{activities}\;+\;2(\mathrm{total}\;\mathrm{duration}\;\mathrm{of}\;\mathrm{vigorous}\;\mathrm{activities})}{\mathrm{maximum}\;\mathrm{number}\;\mathrm{of}\;\mathrm{weeks}’\;\mathrm{usage}\;\mathrm{in}\;12-\mathrm{month}\;\mathrm{data}\;\mathrm{period}}$$

We then classified all weekly moderate/vigorous LCPA user scores by WHO activity category (<30 min/week, 30–149 min/week and ≥150 min/week) [35].

We examined descriptive participant characteristics for all PARS users and total usage/usage by main activity type for all PARS users, PARS participants and PARS completers. We also examined average number of attendances, average length of usage (based on the maximum number of weeks usage data field), weekly moderate/vigorous LCPA user scores, and categories of PA. Finally, we examined associations of demographic variables with PA categories achieved.

#### Statistical Analysis

We performed all statistical analyses using SPSS V26 (IBM, New York, NY, USA). We analysed baseline characteristics for PARS users using the Pearson χ^2^ test for categorical variables (summarized descriptively as frequencies/percentages). After examining data distribution for total attendance using the Kolmogorov–Smirnov test, we calculated median usage periods and LCPA scores for PARS participants and PARS completers. A Mann–Whitney U test using an exact sampling distribution for U [36] determined whether there were differences in number of attendances and weekly LCPA between PARS participants and PARS completers. We used ordinal regression to evaluate the association between demographic variables (sex, age groups and IMD quintiles), and categorical weekly LCPA (<30 min/week, 30–149 min/week and ≥150 min/week) for all PARS users. We then stratified subgroup analyses by PARS stage (PARS participant or completer). The proportional odds assumption for ordinal regression models were tested and not violated. We reported odds ratios (ORs) with 95% confidence intervals and calculated two-sided *p* values for all tests, with *p* < 0.05 considered significant.

## 3. Results

### 3.1. Participant Characteristics

In total, 1194 registered PARS users (448 PARS participants and 746 PARS completers) attended the leisure centres during the data period (1 March 2019–29 February 2020). There were significant differences in sex (χ^2^ (1) = 6.683, *p* = 0.010), age group (χ^2^ (3) = 63.995, *p* < 0.001) and IMD quintile (χ^2^ (4) = 20.467, *p* < 0.001) between PARS participants and PARS completers. PARS participants were more likely to be female (57.6% compared to 50.0% of PARS completers) and younger (33.9% were <60 years compared to 14.4% of PARS completers). PARS completers were more likely to be from the most deprived IMD quintile (23.1% compared to 12.9% of PARS participants) (Table 1).

### 3.2. Attendance and Activity Choices

PARS users attended 46,940 activity sessions in the 12-month data extract period. The most popular activity for PARS participants was PARS specific sessions and for completers was PARS specific exit route sessions (55.1% and 60.6%, respectively). The proportion of independent gym usage was similar for PARS participants and completers (21.4% and 21.6%, respectively). Swimming accounted for 9.2% of PARS participant attendance, however only accounted for 3.2% of PARS completer attendance (Table 2).

### 3.3. Attendance and Activity Choices by Sex

The most popular activity for females was PARS sessions (54.9% of female visits). This was consistent for both female PARS participants (50.9% of female visits) and PARS completers (56.0%). The most popular activity for males was also PARS sessions (64.9% of visits). Similar to females, this was consistent for both male PARS participants (62.3% of male visits) and PARS completers (65.2% of male visits). Female users’ second most popular activity was fitness classes (21.2% of female usage), whereas fitness classes only accounted for 4.5% of male usage. Gym visits were the second most popular activity for male users (26.1%) whereas female gym usage was the third most popular activity and contributed 17.4% of overall usage (Table 3).

Females accounted for 52.1% of overall attendances. However, males attended a higher median number of activities per user than females (median 29.0 (IQR 9.8–61.3) vs. 24.0 (IQR 7.0–53.0).

### 3.4. Attendance and Activity Choices by Age Group

The most popular activity for all PARS users across all age ranges was specific PARS sessions. Older participants were more likely to attend specific PARS sessions than those who were younger (45.5% of overall attendance for <50 years, 63.5% for 70+ years). Proportionally, the highest level of gym use was by younger people (30.6% of overall attendance for <50 years, 19.6% for 70+ years). Activity choices by age were similar for both PARS participants and PARS completers (Table 4).

Over 70s accounted for 59.7% of overall attendances. There was no statistical difference in weekly LCPA between age groups (<50 years median 56.9 min/week (IQR 25.2–90.0), 50–59 median 59.3 min/week (IQR 36.8–99.2), 50–59 years median 59.3 min/week (IQR 36.8–99.2), 60–69 years median 66.1 min/week (IQR 40.4–96.4), 70+ years median 62.8 min/week (IQR 35.1–97.3)).

### 3.5. Attendance and Weekly Moderate/Vigorous Leisure Centre-Based Physical Activity

The median number of attendances for PARS participants was 9.0 (IQR 4.0–20.0). PARS completers attended a median of 42.0 times (IQR 21.8–72.3). Median weekly moderate/vigorous LCPA was significantly different between PARS participants (57.0 (IQR 26.0–90.0) min/week) and PARS completers (68.0 (IQR 42.0–100.0) min/week), (U = 196,285.5, z = 5.058, *p* < 0.001) (Table 5).

Only 6.0% of PARS participants and 9.4% of PARS completers achieved the WHO recommended levels of PA via LCPA per week (Table 6). However, 72.7% of users achieved at least 30 min of PA per week towards the WHO guidelines.

### 3.6. Predictors of Moderate/Vigorous Leisure Centre-Based Physical Activity

Using <30 min per week as the reference category, PARS completers were more likely to achieve a higher level of physical activity category (30–149 min per week or ≥150 min per week) compared to PARS participants (OR 2.14, 95% CI 1.61–2.82, *p* < 0.001). Female PARS participants were less likely to achieve a higher level of physical activity category when compared to male PARS participants (OR 0.65 95% CI 0.43–0.97, *p* < 0.05). PARS participants from the most deprived quintile had significantly increased odds of achieving a higher level of physical activity category when compared to PARS participants from the least deprived quintile (OR 2.37, 95% CI 1.19–4.74, *p* < 0.05) (Table 7).

## 4. Discussion

Our study demonstrated that during the PARS, participants took part in a median of 57.0 min (IQR 26.0–90.0) of moderate/vigorous LCPA per week, and those who continued to attend leisure centre activities after completion of the scheme took part in 68.0 min (IQR 42.0–100.0) of moderate/vigorous LCPA. This means that PARS participants achieve 38.0% and PARS completers 45.3% of the WHO recommended ≥150 min of moderate/vigorous weekly PA [8] through leisure centre use. This highlights the importance of promoting additional PA outside supervised sessions or encouraging participants to attend more sessions per week in order to ensure that participants achieve WHO recommended PA levels.

The majority of usage (59.7%) was at specific PARS sessions, although independent attendance at the gym (21.6% of usage) and fitness classes (13.2% of usage) indicated that the PARS examined did not operate on a ‘one size fits all’ basis, and that participants welcomed other options than PARS supervised sessions. Our findings indicate that PARS completers were more likely to achieve a higher level of physical activity category when compared with PARS participants. Female PARS participants were less likely to achieve a higher level of physical activity category when compared to male PARS participants. PARS participants from the most deprived quintile were more likely to achieve a higher level of physical activity category when compared to PARS participants from the least deprived quintile.

### 4.1. Leisure Centre-Based Physical Activity

A novel finding of this study is how much (57.0 min (IQR 26.0–90.0) moderate/vigorous LCPA per week and what type of activities (mainly supervised PARS sessions, gym and fitness classes) PARS participants attended in leisure centres during the active scheme period. PARS completers who continued to use the leisure centres after finishing the scheme became regular attenders (median attendance for completers in the 12-month period was 42.0 (IQR 21.8–72.3) times. This group had significantly higher weekly LCPA compared to those currently participating in the PARS and were more than twice as likely achieve a higher level of physical activity category when compared to PARS participants. Most existing PARS studies have used self-report PA measures to assess PA behaviour pre and post-scheme [14,23,24,25,26,27,28,29,30] rather than activity undertaken during the scheme. Such studies are not directly comparable with findings as they report all PA, rather than just that undertaken during scheme participation. As with our study, however, they suggest many participants achieve moderate levels of PA levels post-scheme (e.g., 48% of participants achieving 90+ min at 6 months, with 40.2% maintaining this at 12-months [29]).

Previous studies of the Northumberland PARS have reported weekly moderate PA levels of 81 min per week at 24-weeks for all referral conditions [26] and moderate levels of PA at 24 and 52 weeks (based on Godin health contribution scores) [37,38] for participants referred for excess weight [30]. The current study contributes new understanding of how much of this PA takes place in a leisure centre environment (approximately 70%) and what type of activities are most popular. As with the e-coachER trial, very few participants in our study achieved ≥150 min of PA (6.0% after 4 months, versus 6.0% during the 24-week PARS in our study) [39]. However, our results suggest that PARS exit route sessions are an important element to help completers maintain or increase PA levels following PARS, with 9.4% of PARS completers achieving this via leisure centre use in our study. In addition, our study is likely to underestimate weekly PA, as it does not account for any activity undertaken outside the centres. We are not aware of any studies that have examined how much PA is undertaken by PARS participants in leisure facilities compared with activities of daily living or independent PA choices such as walking. However, Rowley and colleagues recently reported that walking contributed 54 min per week of self-reported PA for PARS completers [40]. If this is similar for the PARS participants in this study, there is still a need to promote further PA to reach the WHO recommended levels.

### 4.2. Demographic Differences in Leisure Centre-Based Physical Activity

This study reported that PARS participants from the most deprived IMD quintile were more likely to achieve a higher level of physical activity category when compared to PARS participants living in the least deprived quintile. Although only 12.9% of PARS participants were from the most deprived IMD, this increased to 23.1% of PARS completers, suggesting that when this PARS successfully engaged with those from deprived areas, they were more likely to continue to use the leisure centres following completion. PARS are therefore likely to be a useful intervention for those living in more deprived areas if these referrals can be encouraged to engage and adhere. This is important as we have previously reported that those from more deprived areas are less likely to attend local authority leisure centres [33] and that greater deprivation was a negative predictor of 12-week adherence to PARS in Northumberland [26]. Analysis of the National Exercise Referral Scheme in Wales provides evidence that PARS can engage across the full spectrum of socioeconomic status’ [41] and unlike analysis from Northumberland, other studies have reported no significant differences in adherence between socioeconomic groups [42,43]. Systematic review evidence exploring adherence to PARS suggests that cost is a barrier to participation during the scheme period but also post PARS, when often schemes are no longer subsidised [44]. We were unable to examine the effect of cost as we did not receive information about membership type (direct debit or payment for each session on attendance) or participant entitlement to reduced price usage from the leisure trust. Further research is required to understand the relationship between cost of PARS and participant adherence and how this may impact PA achieved as part of a scheme.

Our results indicate that female PARS participants were less likely to achieve a higher level of physical activity category compared to male PARS participants. Most studies investigating PARS have focused on gender differences in relation to uptake and adherence. Further studies are required to better understand the relationship between participant characteristics and amount of PA achieved during PARS. A greater proportion of females (57.6%) were classified as PARS participants compared to males (42.2%). This appears to be consistent with other studies indicating that female referral levels are higher than for males [17,23,24,25,26,42,43,45,46]. However, the proportion of male and female PARS completers was equal, suggesting if males successfully engage with PARS they are more likely to continue taking part in LCPA after completion. Although this is not directly comparable with PARS adherence, other studies have reported that males are more likely to adhere [17,25,42]. Further research is required to understand why this is the case.

Our analysis highlights the importance of PARS sessions for both genders and is in keeping with other findings that support from providers and other attendees are important facilitators of PARS adherence [44]. It is therefore important that PARS continue to offer an option for specific supervised sessions. Due to data setup within the FDS, our study was unable to identify the content of specific health referral sessions attended meaning that studies should seek to gain insight into activity preferences within supervised sessions. Where participants chose independent exercise options, males preferred to use the gym, while females preferred to attend group-based fitness classes. This is consistent with wider leisure centre usage [33,47] and findings that females are more likely to be motivated to exercise by spending time with others and meeting friends [48]. Since 45.1% of female and 35.1% of male usage were independent activity options, PARS should ensure that these are discussed with participants, rather than taking a ‘one size fits all’ approach of offering supervised sessions only.

Our results indicate that specific PARS sessions and PARS exit route sessions were the most popular activity across all age ranges. The opportunity to attend structured exercise sessions under the supervision of qualified exercise referral professionals is consistently identified as a primary facilitator of PARS attendance [24,49]. However, gym-based sessions were a more popular choice of independent activity for younger PARS users (30.6% of overall attendance for under 50′s compared to 19.6% for those aged 70+). One possible explanation for the higher proportional attendance at independent gym sessions for younger PARS users is that these individuals are more likely to be in employment and have limited options to attend predominantly daytime PARS sessions. Another possibility is that younger participants experience difficulty assimilating with the PARS social environment of predominantly older adults. This has been postulated to impact on lower PARS adherence and completer rates in younger participants [49] and highlights the importance of PARS staff discussing options with participants and tailoring activity to individuals. Future research should examine the effect of age and employment status on activity choices.

A consistent finding in PARS research is that increasing age is a significant predictor of uptake [41] and adherence [13]. Our study is novel in that it examined which referral sub-groups were most likely to undertake higher levels of LCPA (PARS participants aged 60–69 years were more likely to achieve a higher level of physical activity category compared with PARS participants aged below 50). Although this is not directly comparable with studies measuring adherence, the Northumberland PARS has previously reported increased engagement and adherence for those over 55 years [26]. Since estimates suggest that 31% of the Northumberland population will be over 65 by 2031 [50], PARS involving supervised support are likely to form part of strategies to promote PA engagement in older adults. Further research is required to understand why this PARS appears to be most successful increasing PA for those aged 60–69 year and how this can be improved for other age groups.

### 4.3. Strengths and Limitations

A criticism of previous research is a lack of understanding about the type of PA achieved following PARS. This study contributes to the understanding of the type and the amount of PA achieved following PARS completion in individuals who continue to attend leisure centres. The importance of this study lies in its potential to help identify a clear policy direction for regional level PA promotion that would make a meaningful contribution to overall PA levels. A strength of this analysis is the novel approach to calculating LCPA from a data source not usually used in PARS research. Using individual level data of attendance combined with intensity levels for activities attended provides a more robust analysis than self-reported surveys as it does not involve participant recall. In addition, issues surrounding data collection, participant burden and response bias are avoided. However, measuring attendance using the FDS data may be subject to error as users may not swipe their card to record an activity when entering a facility. Additionally, they may choose to do another activity while onsite without booking, may leave an activity early or may book online and then decide not to attend the activity. We were not able to quantify this although the leisure trust monitored online booking with attendance at sessions and limited booking privileges in the case of repeated non-attendance.

The 448 registered PARS users did not equate to the number of referrals that started the scheme during the data period, as the subscription was time-limited to 24 weeks. The scheme had monitored usage of PARS completers by adding a completer subscription since 2015. Based on Hanson et al., (2013) [26], we predicted that there would be approximately 620 completers per year, meaning that there was potential for 2480 people to have a completer subscription at the point of data download. It is unclear whether the lower number of PARS completers who used the centre in the year examined was due to inefficient processes in the manual addition of the PARS completer subscription, or whether the drop off in leisure-centre usage at scheme end was large. In order to understand this better, a cohort study, with a series of time interrupted data extracts would give more insight.

### 4.4. Implications of This Study

Leisure centre provision in Northumberland accounted for PARS participants achieving 57.0 min (IQR 26.0–90.0) and PARS completers achieving 68.0 min (IQR 42.0–100.0) of the recommended ≥150 min of moderate/vigorous PA per week. WHO guidelines suggest that all adults should undertake regular PA and that doing some PA is better than none [8]. PARS therefore make a valuable contribution to achieving these guidelines but the study illustrates that in the case of this PARS, attending leisure centre activities alone is unlikely to achieve recommended levels of PA. Therefore, PARS require strategies to promote PA outside formal PARS sessions and other leisure centre activities to help participants to achieve WHO guidelines [8]. Further research is required exploring potential solutions to increase leisure-based PA as part of PARS.

## 5. Conclusions

This study presents a novel way to assess the amount of PA achieved by PARS participants and completers. Our results demonstrated PARS participants achieved 38.0% and PARS completers 45.3% of the WHO recommended ≥150 min of moderate/vigorous weekly PA through leisure centre use. We recommend that PARS encourage participants to continue to attend leisure centres following scheme completion, as these users achieved higher levels of weekly activity than those still attending the scheme. Particular focus is required to encourage women and those aged under 60 years to continue to attend post-scheme. We recommend that other providers make use of routinely collected data to examine actual PA behaviour during PARS. We suggest that supervised sessions are an important part of PARS delivery, but schemes should also consider how to encourage independent activity choices where appropriate and account for age and gender.

## Figures and Tables

**Table 1 ijerph-18-02957-t001:** Participant characteristics of PARS users between March 2019 and February 2020.

Participant Characteristic	All Users (n = 1194)	PARS * Participants (n = 448)	PARS * Completers (n = 746)	χ^2^	*p*
	n/%	n/%	n/%		
Sex					
Male	562 (47.1)	190 (42.4)	373 (50)	6.683	0.01
Female	631 (52.9)	258 (57.6)	373 (50)		
Age group					
<50 years	120 (10.0)	69 (15.4)	74 (6.9)	63.995	<0.001
50–59 years	139 (11.6)	83 (18.5)	56 (7.5)		
60–69 years	392 (32.8)	130 (29.0)	262 (35.1)		
70+ years	543 (45.5)	166 (37.1)	377 (50.5)		
IMD **					
IMD1	230 (19.3)	58 (12.9)	172 (23.1)	20.467	<0.001
IMD2	195 (16.3)	75 (16.7)	120 (16.1)		
IMD3	235 (19.7)	102 (22.8)	133 (17.8)		
IMD4	209 (17.5)	77 (17.2)	132 (17.7)		
IMD5	309 (25.9)	128 (28.6)	181 (24.3)		
Not Stated	16 (1.3)	8 (1.8)	8 (1.0)		

PARS *: physical activity referral scheme, IMD **: Index of multiple deprivation (IMD1 = most deprived quintile).

**Table 2 ijerph-18-02957-t002:** Attendance and activity choices.

Activity Type	All Participants (n = 1194)	PARS Participants (n = 448)	PARS Completers (n = 746)
	Attendances (Times, %)	Attendances (Times, %)	Attendances (Times, %)
Gym	10,117 (21.6)	1579 (21.4)	8538 (21.6)
Fitness Classes	6027 (13.2)	954 (12.9)	5253 (13.3)
Swimming	1937 (4.1)	682 (9.2)	1255 (3.2)
Other	655 (1.4)	101 (1.4)	554 (1.4)
PARS/PARS exit route session	28,024 (59.7)	4065 (55.1)	23,959 (60.5)
Total	46,940	7381	39,559

**Table 3 ijerph-18-02957-t003:** Attendance and activity choices by sex.

Activity Type	Female	Male
	Attendances (% of Usage)	Attendances (% of Usage)
All participants (n = 1193)	(n = 631)	(n = 562)
Gym	4248 (17.4)	5869 (26.1)
Fitness Classes	5189 (21.2)	1018 (4.5)
Swimming	1238 (5.1)	699 (3.1)
Other	346 (1.4)	309 (1.4)
PARS/PARS exit route session	13,444 (54.9)	14,579 (64.9)
Total	24,465	22,474
PARS participants (n = 447)	(n = 258)	(n = 189)
Gym	733 (16.1)	846 (30.0)
Fitness Classes	881 (19.3)	73 (2.6)
Swimming	612 (13.4)	70 (2.5)
Other	26 (0.6)	75 (2.6)
PARS/PARS exit route session	2305 (50.6)	1759 (62.3)
Total	4557	2823
PARS completers (n = 746)	(n = 373)	(n = 373
Gym	3515 (17.7)	5023 (25.6)
Fitness Classes	4308 (21.6)	945 (4.8)
Swimming	626 (3.1)	629 (3.2)
Other	320 (1.6)	234 (1.2)
PARS/PARS exit route session	11,139 (56.0)	12,820 (65.2)
Total	19,908	19,651

**Table 4 ijerph-18-02957-t004:** Attendance and activity choices by age group.

Activity Type	<50 Years	50–59 Years	60–69 Years	70+ Years
	Attendances (Times, %)	Attendances (Times, %)	Attendances (Times, %)	Attendances (Times, %)
All users (n = 1194)				
Gym	1219 (30.6)	1064 (22.8)	3579 (21.6)	4255 (19.6)
Fitness Classes	929 (23.3)	527 (11.2)	2399 (14.5)	2352 (10.8)
Swimming	124 (3.1)	358 (7.7)	901 (5.4)	554 (2.5)
Other	3 (0.1)	152 (3.3)	286 (1.7)	214 (0.9)
PARS/PARS exit route session	1707 (42.9)	2563 (55.0)	9399 (56.7)	14,355 (66.1)
Total attendances	3982	4664	16,564	21,730
PARS participants (448)				
Gym	319 (32.8)	292 (20.8)	494 (20.1)	474 (18.6)
Fitness Classes	182 (18.7)	90 (6.4)	449 (18.3)	233 (9.1)
Swimming	29 (3.0)	290 (20.7)	154 (6.3)	209 (8.2)
Other	0 (0.0)	74 (5.3)	11 (0.4)	16 (0.6)
PARS session	442 (45.5)	655 (46.8)	1349 (54.9)	1619 (63.5)
Total attendances	972	1401	2457	2551
PARS completers (746)				
Gym	900 (29.9)	772 (23.7)	3085 (21.9)	3781 (19.7)
Fitness Classes	747 (24.8)	437 (13.4)	1950 (13.8)	2119 (11.0)
Swimming	95 (3.2)	68 (2.1)	747 (5.3)	345 (1.8)
Other	3 (0.01)	78 (2.4)	275 (1.9)	198 (1.0)
PARS exit route session	1265 (42.0)	1908 (58.5)	8050 (57.1)	12,736 (66.4)
Total attendances	3010	3263	14,107	19,179

**Table 5 ijerph-18-02957-t005:** Overall attendance and actual weeks usage.

Level of Activity	All Participants (n = 1194)	PARS Participants (n = 448)	PARS Completers (n = 746)	*p*
	Median (IQR)	Median (IQR)	Median (IQR)	
Number of Attendances	26.0 (8.0–58.0)	9.0 (4.0–20.0)	42.0 (21.8–72.3)	
Actual Weeks Usage	36.0 (14.0–51.0)	13.0 (5.0–25.8)	49.0 (32.8–51.0)	
Attendances per week	1.1 (0.6–1.6)	1.0 (0.5–1.6)	1.1 (0.7–1.6)	0.1
Weekly moderate/vigorous LCPA (min)	64.0 (36.0–96.0)	57.0 (26.0–90.0)	68.0 (42.0–100.0)	<0.001

IQR: interquartile range, LCPA: leisure centre-based physical activity.

**Table 6 ijerph-18-02957-t006:** Weekly moderate/vigorous leisure centre-based physical activity.

Activity Category	All Users		PARS Participants		PARS Completers	
	n	%	n	%	n	%
<30 min per week	229	19.2	128	28.6	101	13.5
30–149 min per week	868	72.7	293	65.4	575	77.1
≥150 min per week	97	8.1	27	6.0	70	9.4

**Table 7 ijerph-18-02957-t007:** Odds ratios of achieving higher categorical levels of weekly leisure centre-based physical activity by demographic variables.

Characteristic	All Participants	PARS Participants	PARS Completers
	OR (95% CI)	OR (95% CI)	OR (95% CI)
Membership type			
PARS participant	1		
PARS completer	2.14 (1.61–2.82) **		
Sex			
Male	1	1	1
Female	0.90 (0.70–1.17)	0.65 (0.43–0.97) *	1.16 (0.82–1.63)
Age group			
<50 years	1	1	1
50–59 years	1.38 (0.81–2.36)	1.26 (0.65–2.45)	1.65 (0.67–4.08)
60–69 years	1.52 (0.96–2.40)	2.08 (1.11–3.91) *	1.20 (0.59–2.44)
70+ years	1.24 (0.80–1.93)	1.21 (0.67–2.19)	1.19 (0.62–2.39)
IMD ^ quintile			
Least deprived 20%	1	1	1
21–40%	1.21 (0.82–1.81)	1.37 (0.75–2.52)	1.11 (0.65–1.90)
41–60%	0.87 (0.60–1.27)	1.01 (0.59–1.74)	0.72 (4.42–1.23)
61–80%	0.89 (0.60–1.33)	0.88 (0.48–1.59)	0.88 (0.51–1.53)
Most deprived 20%	1.71 (1.15–2.55) *	2.37 (1.19–4.74) *	1.47 (0.89–2.44)

* < 0.05, ** < 0.000, ^ Index of multiple deprivation.

## Data Availability

Restrictions apply to the availability of these data. Data were obtained from Active Northumberland and are available from jordan.bell@napier.ac.uk with the permission of Active Northumberland.

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
