# Peer review of "Understanding Leisure Centre-Based Physical Activity after Physical Activity Referral: Evidence from Scheme Participants and Completers in Northumberland UK"

_ijerph, 2021, doi:10.3390/ijerph18062957_

Round 1

Reviewer 1 Report

Dear authors,
congratulations for your work. Here are my considerations:

i) you should do the statistic based on the age of the participants and their group. You can not make conclusions based on a sample with a enormous age range. A person with less than 50 years is completely different physiologically from a 70 years old person. The activity choices may be different if you take into account the age of the participants. Re-Write your discussion with this new results.

ii) Your sample should have exclusion criteria. Example: disability or bone injury.

iii)your study shows that we still bellow the WHO recommendations and more 

Reviewer 2 Report

I did appreciate reading: Understanding leisure centre-based physical activity after physical activity referral: evidence from scheme participants and completers in Northumberland UK.

The authors give an nice insight in PA based on objective data.

I have only one suggestion for improvement. The authors themselves mention that PA might be underestimated since the measured PA does not account for any activity undertaken outside the centres. It would be nice if the authors try to say more about this underestimation. Are there no studies that give an idea about the relative contribution of PA in daily live compared to prescribed or explicit exercise programmes. Other studies (even) relying on selfreported data might give an idea. This underestimation is especially relevant to get an idea how much effort is needed to achieve the WHO guidelines.

Reviewer 3 Report

Some comments are listed below: Please add some sentences on rationale of significance of this study, as you wished to address the exsitant literature gaps in the introduction section, like Line 66-68. How did you distinguish PARS participants and PARS completers, what was the difference? Could you define it where necessary in the method section. Please definte the IQR where first appearance. The discussion was comprehensive and based on the results, which had insightful analysis. Overall, I think it was a good paper that needs some minor revisions.

Reviewer 4 Report

This is a very interesting study which shows long-term monitoring data. It was not certainly logistically and organizationally simple to undertake the monitoring and obtain the data. The frequency of monitoring participants is sizable. The authors tried to take into account a number of variables during the monitoring an exercise program. Evaluation of the frequency of people participation and time participation in a variety of activities is one of the well accessible processes for evaluating parameters of physical activities. The question remains how to record this basic data. The authors declare maximal objectivity of these processes even by using recording devices. The problem with this kind of evaluation process is usually the involvement of energy intensity of participation in selected physical activities. Therefore, it is a surprise to me, that the authors did not proceed in the evaluation through MET (min) presentation. It comes to me even as a bigger surprise because in the text there are mentioned these units and they are even presented in a total of five areas of the evaluated activities. Within their calculations and results they rely primarily on the frequency and time amount of participation in the physical activities. This is alright, of course. However, they provide slightly different picture of the evaluation parameters of physical activities. The question is whether to indicate the calculations in the form of MET. If so, it would be appropriate to significantly link this parameter with the presented results. The authors determined exercise class a MET value 5.0 METs and they classified it as moderate activity. With the exception of gym and swimming they rely on the timetable durations. I have to admit, that I lack connection of data in METs and results. However, the authors continue to operate correctly with the obtained data and also draw attention to the possible effects of age differences in the evaluated groups. Many of the presented results are based on determining the frequency of participation and their duration. The use of median numbers (IQR) is suitable for these calculations. I find the statistical part to be correct. This is an adequate choice for the evaluation of structurally relatively heterogeneous groups.

The discussion is accurate and comprehensive, and it correctly states the advantages and limits of the study. After all, the limits of this study are obvious, even record of participation in particular activities and collection of the data can raise questions. In my opinion, employment issues of people and thus the possibility to participate in the activities of the centre should be also included in the discussion. This should be due to the fact that there is an age difference between the PARS participants and PARS completers. Age-related differences in representation can also raise questions about activities besides the centre. These pieces of information should be more emphasized in the discussion.

Withing the conclusions, I would welcome a brief evaluation of selected results, even in structured form. I assume, that the authors unnecessarily limit themselves to a mere presentation in the form of some kind of recommendation.

What is the reason for such significantly different data in the Table 4 in parameters: Number of Attendances and Actual Weeks Usage?

Formal adjustments

Table 3. Attendance and activity choices by sex – is this table compatible with graphical norms for publishing?

Table 3. The total amount of percentages in the first, fourth and fifth column is not 100%. Take into consideration some appropriate adjustments of the individual percentages.  

Row 225 – Marking [36] is repeating

Row markings are projected into Table 6 – make corrections for the final version

I will recommend the study for publishing if my comments are taken into consideration and if there is a reaction to my comments.

This is a worthy work, which can provide feedback on the success of exercise programs for the adult population.
